# GENERATING VALID EUCLIDEAN DISTANCE MATRICES

## ABSTRACT

Generating point clouds, e.g., molecular structures, in arbitrary rotations, translations, and enumerations remains a challenging task. Meanwhile, neural networks utilizing symmetry invariant layers have been shown to be able to optimize their training objective in a data-efficient way. In this spirit, we present an architecture which allows to produce valid Euclidean distance matrices, which by construction are already invariant under rotation and translation of the described object. Motivated by the goal to generate molecular structures in Cartesian space, we use this architecture to construct a Wasserstein GAN utilizing a permutation invariant critic network. This makes it possible to generate molecular structures in a one-shot fashion by producing Euclidean distance matrices which have a three-dimensional embedding.

## 1 INTRODUCTION

Recently there has been great interest in deep learning based on graph structures (Defferrard et al., 2016; Kipf & Welling, 2016; Gilmer et al., 2017) and point clouds (Qi et al., 2017; Li et al., 2018b; Yang et al., 2019). A prominent application example is that of molecules, for which both inference based on the chemical compound, i.e., the molecular graph structure (Kearnes et al., 2016; Janet & Kulik, 2017; Winter et al., 2019a), and based on the geometry, i.e. the positions of atoms in 3D space (Behler & Parrinello, 2007; Rupp et al., 2012; Schütt et al., 2017a; Smith et al., 2017) are active areas of research.

A particularly interesting branch of machine learning for molecules is the reverse problem of generating molecular structures, as it opens the door for designing molecules, e.g., obtain new materials (Sanchez-Lengeling & Aspuru-Guzik, 2018; Barnes et al., 2018; Elton et al., 2018; Li et al., 2018a), design or discover pharmacological molecules such as inhibitors or antibodies (Popova et al., 2018; Griffen et al., 2018), optimize biotechnological processes (Guimaraes et al., 2017). While this area of research has exploded in the past few years, the vast body of work has been done on the generation of new molecular compounds, i.e. the search for new molecular graphs, based on string encodings of that graph structure or other representations (Gómez-Bombarelli et al., 2018; Winter et al., 2019b). On the other hand, exploring the geometry space of the individual chemical compound is equally important, as the molecular geometries and their probabilities determine all equilibrium properties, such as binding affinity, solubility etc. Sampling different geometric structures is, however, still largely left to molecular dynamics (MD) simulation that suffers from the rare event sampling problem, although recently machine learning has been used to speed up MD simulation (Ribeiro et al., 2018; Bonati et al., 2019; Zhang et al., 2019; Plattner et al., 2017; Doerr & Fabritiis, 2014) or to perform sampling of the equilibrium distribution directly, without MD (Noé et al., 2019). All of these techniques only sample one single chemical compound in geometry space.

Here we explore—to our best knowledge for the first time in depth—the simultaneous generation of chemical compounds and geometries. The only related work we are aware of (Gebauer et al., 2018; 2019) demonstrates the generation of chemical compounds, placing atom by atom with an autoregressive model. It was shown that the model can recover compounds contained in the QM9 database of small molecules (Ruddigkeit et al., 2012; Ramakrishnan et al., 2014) when trained on a subset, but different configurations of the same molecule were not analyzed.

While autoregressive models seem to work well in the case of small ($< 9$ heavy atoms) molecules like the ones in the QM9 database, they can be tricky for larger structures as the probability to completely form complex structures, such as rings, decays with the number of involved steps.

To avoid this limitation, in our method we investigate in one shot models for point clouds which have no absolute orientation, i.e., the point cloud structure is considered to be the same independent of its rotation, translation, and of the permutation of points.

A natural representation, which is independent of rotation and translation is the Euclidean distance matrix, which is the matrix of all squared pairwise Euclidean distances. Furthermore, Euclidean distance matrices are useful determinants of valid molecular structures.

While a symmetric and non-negative matrix with a zero diagonal can easily be parameterized by, e.g., a neural network, it is not immediately clear that this matrix indeed belongs to a set of $n$ points in Euclidean space and even then, that this space has the right dimension.

Here we develop a new method to parameterize and generate valid Euclidean distance matrices without placing coordinates directly, hereby taking away a lot of the ambiguity.

We furthermore propose a Wasserstein GAN architecture for learning distributions of pointclouds, e.g., molecular structures invariant to rotation, translation, and permutation. To this end the data distribution as well as the generator distribution are represented in terms of Euclidean distance matrices.

In summary, our main contributions are as follows:

- We introduce a new method of training neural networks so that their output are Euclidean distance matrices with a predefined embedding dimension.

- We propose a GAN architecture, which can learn distributions of Euclidean distance matrices, while treating the structures described by the distance matrices as set, i.e., invariant under their permutations.

- We apply the proposed architecture to a set of $C_7O_2H_{10}$ isomers contained in the QM9 database and show that it can recover parts of the training set as well as generalize out of it.

## 2 GENERATING EUCLIDEAN DISTANCE MATRICES

We describe a way to generate Euclidean distance matrices $D \in \mathbb{EDM}^n \subset \mathbb{R}^{n \times n}$ without placing coordinates in Cartesian space. This means in particular that the parameterized output is invariant to translation and rotation.

A matrix $D$ is in $\mathbb{EDM}^n$ by definition if there exist points $\boldsymbol{x}_1, \ldots, \boldsymbol{x}_n \in \mathbb{R}^d$ such that $D_{ij} = \|\boldsymbol{x}_i - \boldsymbol{x}_j\|_2^2$ for all $i, j = 1, \ldots, n$. Such a matrix $D$ defines a structure in Euclidean space $\mathbb{R}^d$ up to a combination of translation, rotation, and mirroring. The smallest integer $d > 0$ for which a set of $n$ points in $\mathbb{R}^d$ exists that reproduces the matrix $D$ is called the embedding dimension.

The general idea of the generation process is to produce a hollow (i.e., zeros on the diagonal) symmetric matrix $\tilde{D}$ and then weakly enforce $\tilde{D} \in \mathbb{EDM}^n$ through a term in the loss. It can be shown that

$$\tilde{D} \in \mathbb{EDM}^n \Leftrightarrow -\frac{1}{2} J \tilde{D} J \text{ positive semi-definite,} \tag{1}$$

where $J = \mathbb{I} - \frac{1}{n} \mathbf{1}\mathbf{1}^\top$ and $\mathbf{1} = (1, \ldots, 1)^\top \in \mathbb{R}^n$ (Schoenberg, 1935; Gower, 1982). However trying to use this relationship directly in the context of deep learning by parameterizing the matrix $\tilde{D}$ poses a problem, as the set of EDMs forms a cone (Dattorro (2010)) and not a vector space, which is the underlying assumption of the standard optimizers in common deep learning frameworks. One can either turn to optimization techniques on Riemannian manifolds (Zhang et al. (2016)) or find a reparameterization in which the network's output behaves like a vector space and that can be transformed into an EDM.

Here, we leverage a connection between EDMs and positive semi-definite matrices Alfakih et al. (1999); Krislock & Wolkowicz (2012) in order to parameterize the problem in a space that behaves like a vector space. In particular, for $D \in \mathbb{EDM}^n$ by definition there exist points $\boldsymbol{x}_1, \ldots, \boldsymbol{x}_n \in \mathbb{R}^d$ generating $D$. The EDM $D$ has a corresponding Gram matrix $M \in \mathbb{R}^{n \times n}$ by the relationship

$$M_{ij} = \langle \boldsymbol{y}_i, \boldsymbol{y}_j \rangle_2 = \frac{1}{2}(D_{1j} + D_{i1} - D_{ij}) \tag{2}$$

with $\boldsymbol{y}_k = \boldsymbol{x}_k - \boldsymbol{x}_1,\ k = 1, \ldots, n$ and vice versa

$$D_{ij} = M_{ii} + M_{jj} - 2M_{ij}. \tag{3}$$

The matrix $M$ furthermore has a specific structure

$$M = \begin{pmatrix} 0 & \mathbf{0}^\top \\ \mathbf{0} & L \end{pmatrix} \tag{4}$$

with $L \in \mathbb{R}^{n-1 \times n-1}$ and is symmetric and positive semi-definite. It therefore admits an eigenvalue decomposition $M = USU^\top = (U\sqrt{S})(U\sqrt{S})^\top = YY^\top$ which, assuming that $S = \mathrm{diag}(\lambda_1, \ldots, \lambda_n)$ with $\lambda_1 \geq \lambda_2 \geq \ldots \geq \lambda_n \geq 0$, reveals a composition of coordinates $Y$ in the first $d$ rows where $d$ is the embedding dimension and the number of non-zero eigenvalues of $M$ associated to $D$.

Therefore, the embedding dimension $d$ of $D$ is given by the rank of $M$ or equivalently the number of positive eigenvalues. In principle it would be sufficient to parameterize a symmetric positive semi-definite matrix $L \in \mathbb{R}^{n-1 \times n-1}$, as it then automatically is also a Gram matrix for some set of vectors. However, also the set of symmetric positive semi-definite matrices behaves like a cone, which precludes the use of standard optimization techniques.

Instead, we propose to parameterize an arbitrary symmetric matrix $\tilde{L} \in \mathbb{R}^{n-1 \times n-1}$, as the set of symmetric matrices behaves like a vector space. This symmetric matrix can be transformed into a symmetric positive semi-definite matrix

$$L = g(\tilde{L}) = g\left(U \begin{pmatrix} \lambda_1 & & \\ & \ddots & \\ & & \lambda_{n-1} \end{pmatrix} U^\top\right) = U \begin{pmatrix} g(\lambda_1) & & \\ & \ddots & \\ & & g(\lambda_{n-1}) \end{pmatrix} U^\top \tag{5}$$

by any non-negative function $g(\cdot)$ and then used to reconstruct $D$ via (3) and (4).

This approach is shown in Algorithm 1 for the context of neural networks and the particular choice of $g = \mathrm{sp}$, the softplus activation function. A symmetric matrix $\tilde{L}$ is parameterized and transformed into a Gram matrix $M$ and a matrix $D$. For $M$ there is a loss in place that drives it towards a specific rank and for $D$ we introduce a penalty on negative eigenvalues of (1). It should be noted that $g(\cdot)$ can also be applied for the largest $d$ eigenvalues and explicitly set to 0 otherwise, in which case the rank of $M$ is automatically $\leq d$. In that case it is not necessary to apply $L_{\mathrm{rank}}$.

---

**Algorithm 1** Algorithm to train a generative neural network to (in general non-uniformly) sample Euclidean distance matrices based on the neural network $G$, where $N_z$ is the dimension of the input vector, $m$ the batch size, and $n$ the number of points to place relative to one another.

---

1: Sample $\mathbf{z} \sim \mathcal{N}(0,1)^{m \times N_z}$, i.e., sample from a simple prior distribution,
2: Transform $X = G(\mathbf{z}) \in \mathbb{R}^{m \times (n-1) \times (n-1)}$ via a neural network $G$,
3: **for** $i = 1$ to $m$ **do**
4:     Symmetrize $\tilde{L} \leftarrow \frac{1}{2}\left(X_i + X_i^\top\right)$
5:     Make positive semi-definite $L \leftarrow \mathrm{sp}(\tilde{L})$ with (5)
6:     Assemble $M = M(L)$ with (4)
7:     Assemble $D = D(M)$ with (3)
8:     Compute eigenvalues $\mu_1, \ldots, \mu_n$ of $-\frac{1}{2}JDJ$, see (1)
9:     $L_{\mathrm{edm}}^{(i)} \leftarrow \sum_{k=1}^n \mathrm{ReLU}(-\mu_k)^2$
10:     Compute eigenvalues $\lambda_1, \ldots, \lambda_n$ of $M$ such that $\lambda_1 \geq \lambda_2 \geq \ldots \lambda_n$
11:     $L_{\mathrm{rank}}^{(i)} \leftarrow \sum_{k=d+1}^n \lambda_k^2$
12: **end for**
13: $L \leftarrow \eta_1 \frac{1}{m} \sum_{i=1}^m L_{\mathrm{edm}}^{(i)} + \eta_2 \frac{1}{m} \sum_{i=1}^m L_{\mathrm{rank}}^{(i)}$
14: Optimize weights of $G$ with respect to $\nabla L$.

---

## 3   Euclidean distance matrix WGAN

We consider the class of generative adversarial networks (Goodfellow et al. (2014)) (GANs) and in particular Wasserstein GANs (WGANs), i.e., the ones that minimize the Wasserstein-1 distance

in contrast to the original formulation, where the former can be related to minimizing the Jensen-Shannon divergence Arjovsky et al. (2017). WGANs consist of two networks, one generator network $G(\cdot)$, which transforms a prior distribution—in our case a vector of white noise $\boldsymbol{z} \sim \mathcal{N}(0,1)^{n_z}$—into a target distribution $\mathbb{P}_g$ which should match the data's underlying distribution $\mathbb{P}_r$ as closely as possible. The other network is a so-called critic network $C(\cdot) \in \mathbb{R}$, which assigns scalar values to individual samples from either distribution. High scalar values express that the sample is believed to belong to $\mathbb{P}_r$, low scalar values indicate $\mathbb{P}_g$. The overall optimization objective reads

$$\min_G \max_{C \in \mathcal{D}} \mathbb{E}_{\boldsymbol{x} \sim \mathbb{P}_r} \left[ C(\boldsymbol{x}) \right] - \mathbb{E}_{\boldsymbol{x} \sim \mathbb{P}_g} \left[ C(\boldsymbol{x}) \right], \tag{6}$$

where $\mathcal{D}$ is the set of all Lipschitz continuous functions with a Lipschitz constant $L \leq 1$. We enforce the Lipschitz constant using a gradient penalty (WGAN-GP) introduced in Gulrajani et al. (2017). One can observe that the maximum in Eq. (6) is attained when as large as possible values are assigned to samples from $\mathbb{P}_r$ and as small as possible values to samples from $\mathbb{P}_g$. Meanwhile the minimum over the generator network $G$ tries to minimize that difference, which turns out to be exactly the Wasserstein-1 distance according to the Kantorovich–Rubinstein duality (Villani (2008)). Since the Wasserstein-1 distance is a proper metric of distributions, the generated distribution $\mathbb{P}_g$ is exactly the data distribution $\mathbb{P}_r$ if and only if the maximum in Eq. (6) is zero. The networks $G$ and $C$ are trained in an alternating fashion.

We choose for the critic network the message-passing neural network SchNet (Schütt et al., 2017b;a; 2018) $C_{\mathrm{SchNet}}(\cdot)$, which was originally designed to compute energies of molecules.

It operates on the pairwise distances $(\sqrt{D_{ij}})_{i,j=1}^n$, $D \in \mathbb{EDM}^n$ in a structure and the atom types $\mathcal{T}^n$. If there is no atom type information present, these can be just constant vectors that initially carry no information. These atom types are then embedded into a state vector and transformed with variable sharing across all atoms. Furthermore there are layers in which continuous convolutions are performed based on the relative distances between the atoms. In a physical sense this corresponds to learning energy contributions of, e.g., bonds and angles. Finally all states are mapped to a scalar and then pooled in a sum.

Due to the pooling and the use of only relative distances but never absolute coordinates, the output is invariant under translation, rotation, and permutation.

The generator network $G$ employs the construction of Section 2 to produce approximately EDMs with a fixed embedding dimension. Therefore this architecture is able to learn distributions of Euclidean distance matrices.

# 4 Application and results

The WGAN-GP introduced in Sec. 3 is applied to a subset of the QM9 dataset consisting of 6095 isomers with the chemical formula $C_7O_2H_{10}$. To this end the distribution not only consists of the Euclidean distance matrices describing the molecular structure but also of the atom types. The generator produces an additional type vector in a multi-task fashion which is checked against a constant type reference with a cross-entropy loss. In particular this means that there is another linear transformation between the output of the neural network parameterizing the symmetric matrix $\tilde{L}$ (see, e.g., (5)) and a vector $\mathbf{t} \in \mathbb{R}^{m \times n \times n_{\mathrm{types}}}$ which is due to the use of a softmax non-linearity a probability distribution over the types per atom. This probability distribution is compared against a one-hot encoded type vector $\mathbf{t}_{\mathrm{ref}}$ representing the type composition in the considered isomer with a cross entropy term $H(\mathbf{t}, \mathbf{t}_{\mathrm{ref}})$. As in this example the chemical composition never changes, the type vector $\mathbf{t}_{\mathrm{ref}}$ is always constant.

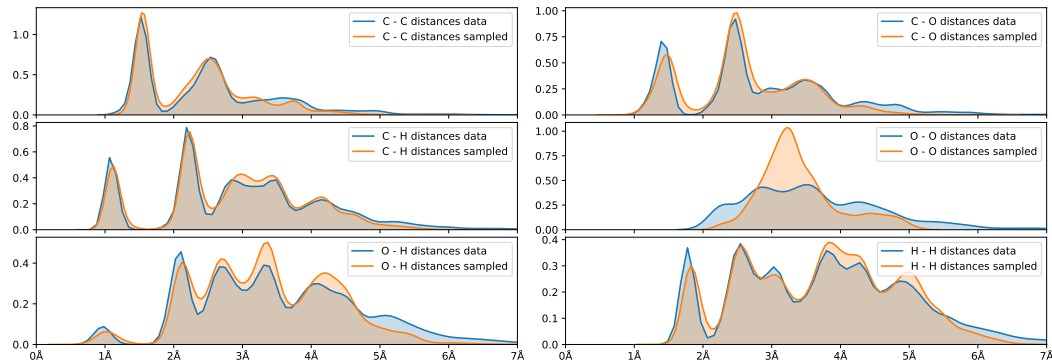

Figure 1: Distribution of pairwise distances between different kinds of atom type after training a Euclidean distance matrix WGAN-GP (Sec. 3) on the $C_7O_2H_{10}$ isomer subset of QM9.

Furthermore the prior of a minimal distance between atoms is applied, i.e., we have a loss penalizing distances that are too small. Altogether we optimize the losses

$$L_{\text{critic}} = \mathbb{E}_{(D,\mathbf{t})\sim\mathbb{P}_g}\left[C(D,\mathbf{t})\right] - \mathbb{E}_{(D,\mathbf{t})\sim\mathbb{P}_r}\left[C(D,\mathbf{t})\right] \quad \text{(original WGAN loss)} \tag{7}$$

$$+ \lambda L_{\text{GP}} \quad \text{(gradient penalty of WGAN-GP)} \tag{8}$$

$$+ \varepsilon_{\text{drift}}\mathbb{E}_{(D,\mathbf{t})\sim\mathbb{P}_r}\left[C(D,\mathbf{t})^2\right] \quad \text{(drift term Karras et al. (2017))} \tag{9}$$

$$L_{\text{gen}} = -\mathbb{E}_{(D,\mathbf{t})\sim\mathbb{P}_g}\left[C(D,\mathbf{t})\right] \quad \text{(original WGAN loss)} \tag{10}$$

$$+ \mathbb{E}_{(D,\mathbf{t})\sim\mathbb{P}_g}\left[H(\mathbf{t},\mathbf{t}_{\text{ref}})\right] \quad \text{(cross entropy for types)} \tag{11}$$

$$+ k \cdot \mathbb{E}_{(D,\mathbf{t})\sim\mathbb{P}_g}\left[\frac{1}{2}\sum_{i\neq j}(\sqrt{D_{ij}} - r)^2\right] \quad \text{(harmonic repulsion)} \tag{12}$$

$$+ L_{\text{edm}} \quad \text{(for EDMs, see Alg. 1)} \tag{13}$$

with $C(\cdot)$ being a SchNet critic, $\mathbf{t}_{\text{ref}}$ a reference type order, $\lambda = 10$, $\varepsilon_{\text{drift}} = 10^{-3}$, $k = 10$, and $r$ being the minimal pairwise distance achieved in the considered QM9 subset. The drift term leads to critic values around 0 for real samples, as otherwise only the relative difference between values for real and fake samples matters. Although in principle the cross entropy loss (11) is not required we found in our experiments that it qualitatively helps convergence. The generator network $G(\cdot)$ uses a combination of deconvolution and dense layers.

The function $g(\cdot)$ ensuring positive semi-definiteness (5) was chosen to be the softplus activation $g = \text{sp}$ for the largest three eigenvalues and we explicitly set all other eigenvalues to zero. This leads to a Gram matrix with exactly the right rank and the constraint does not need to be weakly enforced anymore in the generator's loss.

Prior to training the data was split into 50% training and 50% test data. After training on the training data set we evaluate the distribution of pairwise distances between different types of atoms, see Fig. 1. The overall shape of the distributions is picked up and only the distance between pairs of oxygen atoms are not completely correctly distributed.

After generation we perform a computationally cheap validity test by inferring bonds and bond orders with Open Babel O'Boyle et al. (2011). On the inferred bonding graph we check for connectivity and valency, i.e., if for each atom the number of inferred bonds add up to its respective valency. This leaves us with roughly 7.5% of the generated samples.

For the valid samples we infer canonical SMILES representations which are a fingerprint of the molecule's topology in order to determine how many different molecule types can be produced using the trained generator. Fig. 2 shows the cumulative number of unique SMILES fingerprints when producing roughly 4000 valid samples. It can be observed that the network is able to generalize out of the training set and is able to generate not only topologies which can be found in the test set but also entirely new ones. Nevertheless, while the GAN implementation of our approach generates substantial diversity in configuration space, the diversity in chemical space is limited. This

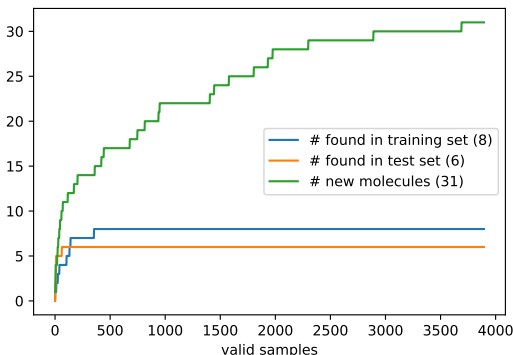

Figure 2: Number of unique molecular structures in terms of their topology for roughly $4000$ valid generated samples and whether they could be found in the training set (blue), the test set (orange), or had a new topology altogether (green).

can be improved by better hyperparameter selection or choosing different generative structures, e.g., (Gebauer et al., 2018; 2019).

While SMILES can be used to get an idea about the different bonding structures that were generated, it contains no information about different possible conformations in these bonding structures. To analyze the number of unique conformations that were generated, we compared each generated structure against all structures in the considered QM9 subset. Since the architecture is designed in such a way that it is permutation invariant, i.e., applying the critic onto a matrix $D = (D_{i,j})_{i,j=1}^{n}$ and $D_\sigma = (D_{\sigma(i),\sigma(j)})_{i,j=1}^{n}$ for some permutation $\sigma$ yields the same result, one first has to find the best possible assignment of atoms.

To this end, we apply the Hungarian algorithm Kuhn (1955) onto a cost matrix $C \in \mathbb{R}^{n \times n}$ for EDMs $D_1, D_2$ and type vectors $\mathbf{t}_1, \mathbf{t}_2 \in \mathbb{R}^n$ with

$$C_{i,j} = \begin{cases} \left| \frac{1}{n} \sum_{k=1}^{n} (D_1)_{i,k} - \frac{1}{n} \sum_{k=1}^{n} (D_2)_{j,k} \right| & \text{, if } (\mathbf{t}_1)_i = (\mathbf{t}_2)_j, \\ \infty & \text{, otherwise.} \end{cases} \tag{14}$$

Intuitively this means that the cost of assigning atom $i$ in the first structure to atom $j$ in the second structure depends on whether their atom types match, in which case we compare the mean distance from the $i$-th atom to all other atoms in its structure to the same quantity for the $j$-th atom in the second structure. If the atom types do not match, we assign a very high number so that this particular mapping is not considered. After we have found an assignment between the atoms, we superpose the structures using functionality from the software package MDTraj (McGibbon et al. (2015)) and evaluate the maximal atomic distance between all heavy atoms (i.e., carbons and oxygens) after alignment. The cutoff at which we consider a structure to be a distinct conformation is a maximal atomic distance between heavy atoms of more than $d_{\text{cutoff}} = 0.6\,\text{Å}$, i.e., more than half a carbon–carbon bond length.

The results of this analysis are depicted in Fig. 3. One can observe that although the reported number of unique molecular structures via SMILES is rather low, under our similarity criterion a lot of different valid conformations are discovered; in particular also some new conformations of structures that were already contained in the QM9 database.

Finally, we also check for the approximate total energies of the generated molecules compared to the database's. To this end, we use the semi-empirical method provided by the software package MOPAC Stewart (1990) to relax all structures in the considered QM9 subset as well as all valid generated structures, see Fig. 4. It can be observed that after relaxation all energies are contained within the same range of roughly $-1586\,\text{eV}$ to $-1581\,\text{eV}$. This means that our architecture is capable to propose structures which after relaxation have energies comparable to the ones in the database.

In Figure 5 we show examples of generated molecules in the top row (A–D) and the closest respective matches in the QM9 database in the bottom row (A'–D'). The closeness of a match was

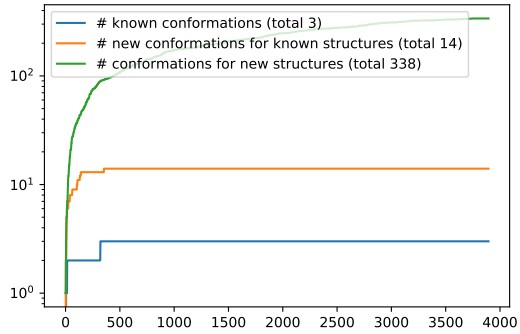

Figure 3: Unique generated conformations up to a maximal heavy atom distance cutoff of $d_{\mathrm{cutoff}} = 0.6\,\text{Å}$ after assignment and superposition. We distinguish the categories of known conformations in the considered subset of the QM9 database (blue), new conformations for contained molecular structures (orange), and distinct conformations for molecular structures that are not contained.

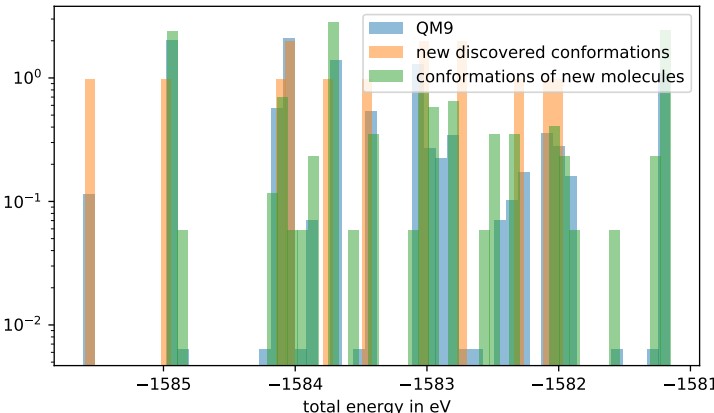

Figure 4: Total energies of structures that were relaxed with the semi-empirical method implemented by MOPAC, in particular for molecules contained in the considered QM9 subset (blue), structures that correspond to new conformations for contained molecules (orange), and unique conformations that belong to new molecules.

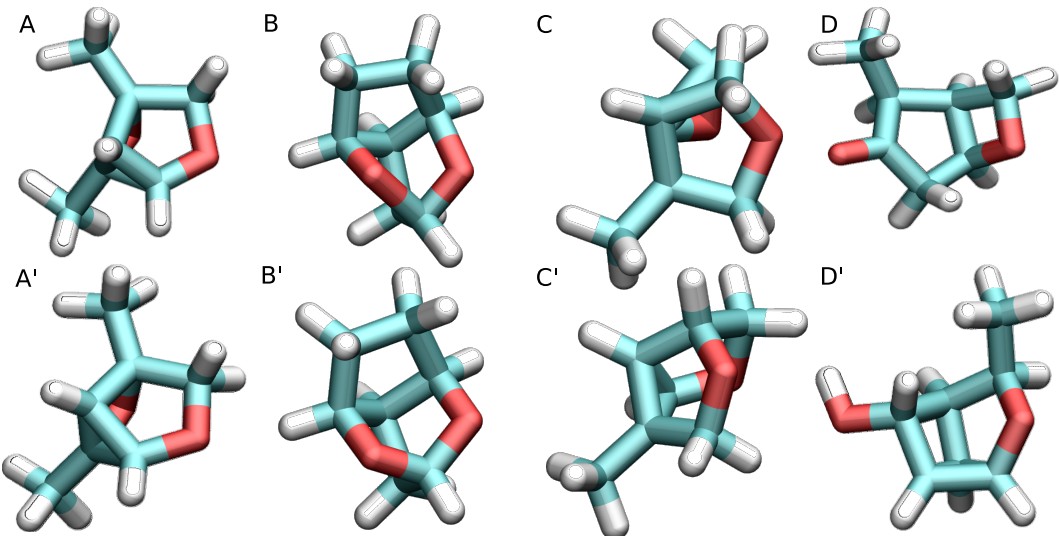

Figure 5: Sampled structures with the Euclidean distance matrix WGAN. Top row A to D are generated samples, bottom row A' to D' are closest matches from the QM9 database. Generated molecules A and B could be matched with A' and B' up to a maximum atom distance of $0.6\,\text{Å}$. Generated molecules C and D are new molecular structures with their closest matches C' and D', respectively.

determined by the maximal atomic distance after assignment of atom identities and superposition. Configurations A and B could be matched with a maximal atomic distance of less than $d_{\text{cutoff}}$.

## 5  CONCLUSION AND DISCUSSION

We have developed a way to parameterize the output of a neural network so that it produces valid Euclidean distance matrices with a predefined embedding dimension without placing coordinates in Cartesian space directly. This enables us to be naturally invariant under rotation and translation of the described object. Given a network that is able to produce valid Euclidean distance matrices we introduce a Wasserstein GAN that can learn to one-shot generate distributions of point clouds irrespective of their orientation, translation, or permutation. The permutation invariance is achieved by incorporating the message passing neural network SchNet as critic.

We applied the introduced WGAN to the $C_7O_2H_{10}$ isomer subset of the QM9 molecules database and could generalize out of the training set as well as achieve a good representation of the distribution of pairwise distances in this set of molecules.

In future work we want to improve on the performance of the network on the isomer subset as well as extend it to molecules of varying size and chemical composition. We expect the ideas of this work to be applicable for, e.g., generating, transforming, coarse graining, or upsampling point clouds.

To improve the quality of the generated molecular structures a follow up of this work would be including a force field so that more energetically reasonable configurations are produced and including penalty terms which favor configurations that do not contain invalid valencies, i.e., produce valid molecular structures. Furthermore optimizing a conditional distribution for a particular molecule—i.e., conditioning on the molecular graph—and exploring its conformations is a natural extension of this work.

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
