# OpenReview forum: "Generating valid Euclidean distance matrices"
_ICLR.cc/2020/Conference — Reject_

### Official Review · AnonReviewer2 · 2019-10-17
**Official Blind Review #2**

**Rating:** 8

**Review:**

The paper is extremely interesting, solid and very well written. The idea is simple but nonetheless developed in a smart and effective fashion. The underlying theory is solid, even if some choices should have been discussed more deeply (e.g. the chosen loss function). Introduction and references are adequate, and the paper is readable by a quite broad audience, despite the detailed technical sections. The main issue related to the manuscript is very narrow target of the experimental part, limited to the isomers of a given compound - it would have been interesting to check its potentialities in generating more different structures and distance matrices, and thus to compare its effectiveness versus alternative generative approaches.

**Experience Assessment:**

I have read many papers in this area.

**Review Assessment: Checking Correctness Of Derivations And Theory:**

I carefully checked the derivations and theory.

**Review Assessment: Checking Correctness Of Experiments:**

I assessed the sensibility of the experiments.

**Review Assessment: Thoroughness In Paper Reading:**

I read the paper at least twice and used my best judgement in assessing the paper.

---

> ### Author Response · Authors · 2019-11-13
> **Response to AnonReviewer2**
>
> Thank you for your kind review and positive feedback!
> - We added to the explanation of the chosen loss function, especially with respect to the atom types via a cross entropy term and added a short explanation of the drift term. Furthermore an intuitive description of attaining high and/or low values in the critic network $C$ was added.
> - Expanding the application scope is indeed something that we plan to do along with bias/penalty terms towards energetically favorable states and non violated valencies.

---

### Official Review · AnonReviewer1 · 2019-10-23
**Official Blind Review #1**

**Rating:** 3

**Review:**

Authors of this paper present architecture to produce valid Euclidean distance matrices. A Wasserstein GAN is then constructed by utilizing a permutation invariant critic network based on the architecture. Generating molecular structures in a one-shot fashion is conducted using the produced distance matrices in 3-d embedding.

In Section 2, the constraint on L makes M symmetric and positive semi-definite. This seems to be equivalent to treating M as a kernel matrix, and D is the pairwise distance between the kernel function induced by M. So, learning a valid Euclidian distance matrix is same as learning a kernel function.

Authors need to provide the evidence that Equation (5) holds for all function g, especially for the used softplus activation function. As I know, it holds if function g is a polynomial matrix function.

In Algorithm 1, what is the meaning of step 14? There is no definition or discussion about G and the \nabla L. The construction function is the key contribution of this paper, which is incorporated the existing SchNet, so it is better to show the advantage of the proposed construction method comparing with SchNet.

Although this paper is an application-oriented paper, the comparisons with baseline methods are preferred, such as some simple and straightforward baselines. Due to the lack of comparisons, it is hard to understand the statement given by authors that “ the current performance… is not optimal and can likely be improved by a better hyperparameter selection”.

In Section 4, authors calculate the similarity of generated molecules by the closest respective matches, which is determined by the maximal atomic distance after assignment of atom identities and superposition. Is this the standard way to compute the similarity between two graph structures?


**Experience Assessment:**

I have read many papers in this area.

**Review Assessment: Checking Correctness Of Derivations And Theory:**

I assessed the sensibility of the derivations and theory.

**Review Assessment: Checking Correctness Of Experiments:**

I assessed the sensibility of the experiments.

**Review Assessment: Thoroughness In Paper Reading:**

I read the paper at least twice and used my best judgement in assessing the paper.

---

> ### Author Response · Authors · 2019-11-13
> **Response to AnonReviewer1**
>
> We thank you for your review, questions, and input. In general however we do not think that the rating of "3 - weak reject" is properly motivated. Addressing the points:
> - We agree that, since the kernel distance is defined as $d(X, Y) = K(X, X) + K(Y, Y) -2K(X, Y)$ for a kernel $K$, producing a valid EDM $D_{ij}=M_{ii}+M_{jj}-2M_{ij}$ (Eq. (3) in the manuscript) is equivalent to learning the symmetric positive semi-definite (kernel) matrix evaluations $M_{ij}$. The rank of $M$ corresponds to the embedding dimension.
> - Equation (5) holds whenever the function $g$ has a preimage space that contains the eigenvalues. There are different ways to define a matrix function, one of them is via the Jacobi canonical form, which reduces in our case to an eigenvalue decomposition and application of the function on the individual eigenvalues (see, e.g., Higham, N. J. (2008). Functions of matrices: theory and computation (Vol. 104). Siam.).
> - Algorithm 1, step 14: It is now noted that G is a neural network. All the steps are differentiable so depending on the optimization objective (for instance in a GAN architecture) backpropagation can yield a target distribution in EDM space (indicated by \nabla L). SchNet in this case is used as a Critic (or in classical GANs as Discriminator) network but is not strictly required to sample in EDM space.
> - Comparison to simple and straightforward baselines is going to be included in future research. While the GAN implementation of our approach generates substantial diversity in configuration space, the diversity in chemical space is limited, but can be improved by choosing different generative structures, e.g., autoregressive (arXiv 1906.00957).
> - The standard way to compare two structures strongly depends on the considered type of graph. In proteins for example there is a polymeric backbone structure that can be leveraged for identity assignment via the help of sequence alignment. Since these are small molecules without polymeric structure and finding the right assignment is in principle NP hard, the Hungarian algorithm is often used to tractably find an optimal solution, where optimal refers to the choice of cost matrix.

---

### Official Review · AnonReviewer3 · 2019-10-23
**Official Blind Review #3**

**Rating:** 8

**Review:**


This paper addresses the important problem of molecular structures generation, and more generally of efficient point-cloud distributions learning in d-dimensional space.
The paper is written clearly, the pseudo-code is presented in a clear way, striking a good balance between explicit-ness and concision.

After my first reading I was disappointed in the experiments, because I read the paper quite quickly at first.  After a second more careful read I understood most of it and was much more enthusiastic. Disclaimer: I am inexperienced in this particular field (GANs and GANs for molecular generation) so I lack literature knowledge, and I may be over-evaluating the quality of performances (compared to recent results).
But currently I believe that the authors are a bit too humble about their results (a rather uncommon phenomenon). There is a part of the method that I would like to have clarified (about bond/atom types), but other than this clarification, I strongly recommend the paper to be accepted.

The paper deals with a couple of distinct, related problems.  One is that of sampling valid (Euclidean) distances matrices (EDM). This is done with algorithm 1.
The paper uses these EDM to train a generator G directly in EDM-space, against a Critic network C (the architecture of which is taken from existing literature).
Some of the output configurations have to be discarded due to incorrect bond types assignment (this is the part that is still obscure to me).
The remaining outputs are chemically sensible in terms of bond types/valence/local chemistry/etc, but also, and quite impressively, have reasonable ground state energies !  In this sense, the paper produces samples that may deserve to be added into the QM9 data set (After some data augmentation using DFT or other physics-validated methods of course).


I have mainly two requests:

1. The bond type or atom assignment is not clearly explained. I ask further detailed questions below, but overall I think you should attempt to make a pedagogical explanation of how the atom types are assigned/learned, and how some of them are later discarded.

'The  generator produces an additional type vector in a multi-task fashion which is checked against a  constant type reference with a cross-entropy loss.'
Could you explain this better ? At least in the appendix. Otherwise this very dense sentence remains mysterious. Eq.11 is currently the single occurrence of the H, t, and tref terms.

About the validity test using Open Babel. Again, what happens with bond types is not very clear. Are they set in stone at generation time, and then most of these affected types are 'wrong' and are discarded by Open Babel ?
Why don't you include this validation at training time (if it's very complicated to do, explain why) ?
In any case, please clarify this paragraph, as for now it is cryptic.

2. Part of the paper goal is to achieve learning of point clouds distributions and not about chemistry.  However, discovery of new structures (and their conformations) is in itself a big topic.  I think it would be good to mention a couple of follow ups to your work in the conclusion. For instance, it would seem rather natural to me to include some equivalent of OpenBAbel and Energy-estimates within the learning loop, so that the generator directly generates valid (open-babel wise) and reasonable (energy level-wise) structures.
Besides the conclusion, you should stress out better the significance of your results with a couple of comments here and there. (I have more precise suggestions below).


Some other comments to improve the paper (in clarity or other):

In algorithm 1, you should precise that G() is your NN-based generator network. At this point of the paper G and z and N_z have never been mentioned. This is a problem when reading the paper for the first time.
Line 5, you could explicitly precise, 'with (5)'.  Also I am puzzled by the use of the softplus here, but later in page 5 you say g() is softplus for the first three eigenvalues, and set to 0 for all others. Why not already anticipate and say this in section 2 ?
Line 8, I would have written 'eigenvalues .... of D'  (what is the role of Eq. 1 here?)

At some point (i.e. around end of section 2), a comment would be welcomed, about whether you may sample the space of EDMs uniformly at random using algorithm 1 (I think you do not, and it is ok, but the question naturally arises and it's not addressed).

´´which transforms a prior distribution into a target distribution´´
you could specify that this prior is the Gaussian N(0,1)^Nz in this case, to better connect with algorithm 1.

About equation 6: it would be nice to have some intuitive explanation of what the output values of C(x) mean. They are scalars that represent the opinion of C on the molecule x, so they are the probability that the observed molecule is 'a true molecule x' ?  You should recall that for the inexperienced reader.

Still about Eq.6 : could you quickly provide the motivation for demanding C to be L-Lipschit with  $L \leq 1$ ?

The drift term Eq.9 seems to be some sort of regularization, maybe it could be mentioned once in the text for completeness?

Using Mopac Stewart and figure 4: you could insist more on this result. Up to that point I was very dubious about the usefulness of the whole work, because I would have expected many generated samples to be highly unrealistic, i.e. having huge energies, and so being extremely unstable. Instead here you show it is not the case, and even all your energies lie within the range of observed energies !  This is a very strong result, that is not obvious to expect, and is highly valuable (even after keeping only 7.5% of structures, this is still a strong result.)
I think this test and the corresponding result should be emphasized more.

You observe new topological types and complain there are not many. I think ~20 new topologies is not a small number (for so few atoms), and you may be rather proud of it.  What is too bad is that you do not have a tool for differentiating between the very similar conformations (that correspond to thermal fluctuations around a given structure) and the conformations that encode a new structure (which does not necessarily means new topology).
If you had this tool, you could enrich figure 3 with a curve showing the new structures (not counting conformational variants).

refs 2017a and 2017b are the same.



**Experience Assessment:**

I do not know much about this area.

**Review Assessment: Checking Correctness Of Derivations And Theory:**

I carefully checked the derivations and theory.

**Review Assessment: Checking Correctness Of Experiments:**

I carefully checked the experiments.

**Review Assessment: Thoroughness In Paper Reading:**

I read the paper thoroughly.

---

> ### Author Response · Authors · 2019-11-13
> **Response to AnonReviewer3**
>
> Thank you for the positive, very detailed, and constructive review. We have adressed your points in the following manner:
> - Main request 1a: The unclear sentence about the type vector and cross entropy is expanded with further details in the manuscript. There is another linear transformation between the output of the neural network parameterizing the symmetric matrix $\tilde{L}$ and a vector $\mathbf{t}\in\mathbb{R}^{m\times n\times n_\mathrm{types}}$ which is due to the use of a softmax non-linearity a probability distribution over the types per atom. This probability distribution is compared against a one-hot encoded type vector $\mathbf{t}_\mathrm{ref}$ representing the type composition in the considered isomer with a cross entropy term $H(\mathbf{t}, \mathbf{t}_\mathrm{ref})$. As in this example the chemical composition never changes, the type vector $\mathbf{t}_\mathrm{ref}$ is always constant.
> - Main request 1b: Bonds and bond types are not generated by the network but inferred by OpenBabel which uses a method based on spatial proximity and atom type for assignment. We discard samples which are disconnected and/or violate valencies based on the inferred bonds. This is something that can potentially be included at least approximately during training by using penalty terms. We consider this a good addition for future work, thank you for the suggestion! A sentence reflecting this (as well as including an energy term) has been added to the conclusion.
> - Main request 2: Indeed adding penalty terms which encourage a valid bonding structure as well as including a force field (in this case for small molecules) should help results and we now mention this in the conclusion.
> - Algorithm 1 now mentions that G is a neural network, z a sample from a simple prior, and that sampling in EDM space is in general not uniform. We added an explanation to the application of the softplus activation in the main text, in particular that one can also apply it for the first $d$ eigenvalues and set it to $0$ for the rest, in which case the rank of $M$ does not need to be optimized for. Which of the two methods (setting to $0$ versus penalizing) is better demands further research. In Line 8 it is not the eigenvalues of $D$ but the eigenvalues of $-0.5JDJ$, which is now mentioned in the text for clarification. It is an if and only if condition on EDMness.
> - In the WGAN section we mention that the prior distribution can be chosen as normal distribution to make the connection to the algorithm and application better/easier.
> - An intuitive explanation for the values of the critic for an individual sample has been added: Low values indicate the belief a sample stems from the generated distribution, high values indicate the belief that a sample stems from the data distribution.
> - The condition on the Lipschitz constant comes from the Kantorovich-Rubinstein duality theorem for the Wasserstein-1 metric. Without it, we are not actually minimizing a metric between distributions. The constant can actually be any positive bound, in which case it is not the Wasserstein-1 metric but a fixed multiple of it, i.e., $L\cdot W^1(p, q)$. Intuitively it is a restriction on the maximal gradient that the critic function can attain, which produces a one-dimensional embedding in which a decision can be made which sample belongs to which distribution.
> - The drift term (Eq. (9)) helps to fixate the critic values for real samples around $0$, otherwise the optimization objective only considers the relative difference; ambiguity is reduced. A short explanation has been added to the main text.
> - We emphasize more on the energies result by an additional sentence in the main text. Also we added that while our GAN implementation has substantial diversity in configuration space and only limited diversity in chemical space, this can be improved by using different generative structures, e.g., (Gebauer et al 2018, 2019).
> - We agree that enriching Figure 3 with new structures by differentiating the very similar ones would improve it and plan to work on that in the future.
> - We fixed the duplicate entry in the bibliography, thanks for spotting that!

---

### Decision · Program_Chairs · 2019-12-19

**Decision:**

Reject

**Comment:**

This paper proposes a parametrisation of Euclidean distance matrices amenable to be used within a differentiable generative model. The resulting model is used in a WGAN architecture and demonstrated empirically in the generation of molecular structures.

Reviewers were positive about the motivation from a specific application area (generation of molecular structures). However, they raised some concerns about the actual significance of the approach. The AC shares these concerns; the methodology essentially amounts to constraining the output of a neural network to be symmetric and positive semidefinite, which is in turn equivalent to producing a non-negative diagonal matrix (corresponding to the eigenvalues). As a result, the AC recommends rejection, and encourages the authors to include simple baselines in the next iteration.